# D²GS: Depth-and-Density Guided Gaussian Splatting for Stable and Accurate Sparse-View Reconstruction

**Meixi Song**[1,2]* **Xin Lin**[3]* **Dizhe Zhang**[2]†‡ **Haodong Li**[3] **Xiangtai Li**[4] **Bo Du**[5] **Lu Qi**[2,5]‡

[1] Tsinghua University  [2] Insta360 Research  [3] University of California, San Diego

[4] Nanyang Technological University  [5] Wuhan University

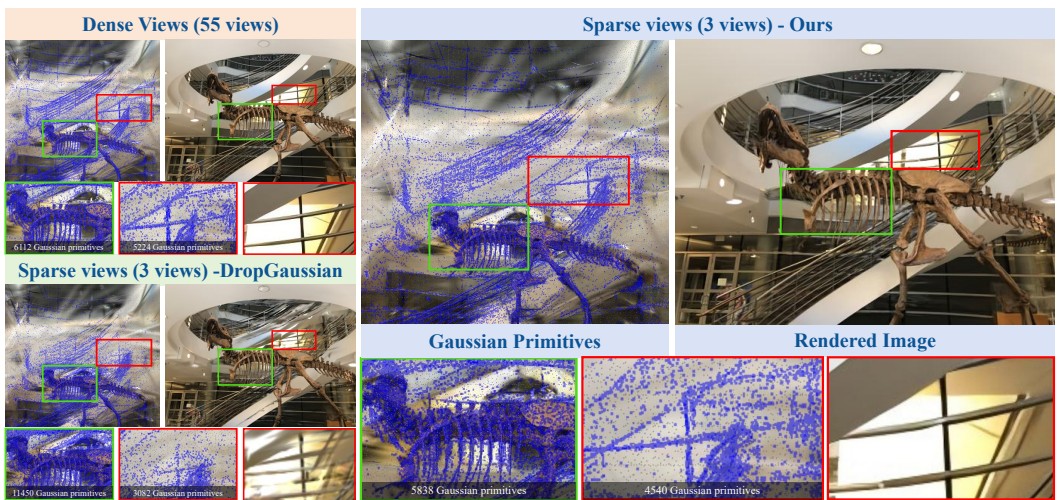

Figure 1: Comparison of Gaussian primitives and rendered images between dense views (55 views) and sparse views (3 views) settings. Overfitting occurs in the near field (green box), while underfitting appears in the far field (red box). The number of Gaussian primitives in the corresponding field is shown below the images.

## Abstract

Recent advances in 3D Gaussian Splatting (3DGS) enable real-time, high-fidelity novel view synthesis (NVS) with explicit 3D representations. However, performance degradation and instability remain significant under sparse-view conditions. In this work, we identify two key failure modes under sparse-view conditions: overfitting in regions with excessive Gaussian density near the camera, and underfitting in distant areas with insufficient Gaussian coverage. To address these challenges, we propose a unified framework D²GS, comprising two key components: a Depth-and-Density Guided Dropout strategy that suppresses overfitting by adaptively masking redundant Gaussians based on density and depth, and a Distance-Aware Fidelity Enhancement module that improves reconstruction quality in under-fitted far-field areas through targeted supervision. Moreover, we introduce a new evaluation metric to quantify the stability of learned Gaussian distributions, providing insights into the robustness of the sparse-view 3DGS. Extensive experiments on multiple datasets demonstrate that our method significantly improves both visual quality and robustness under sparse view conditions. The project page can be found at: https://insta360-research-team.github.io/DDGS-website/.

---

* Equal contribution    † Project lead    ‡ Corresponding author

# 1 INTRODUCTION

Recently, novel view synthesis (NVS) (Kerbl et al., 2023; Lin et al., 2025; Yu et al., 2024; Ye et al., 2024; Lee et al., 2024; Niedermayr et al., 2024; Zhang et al., 2024b; Zhou et al., 2024) and its applications have witnessed significant progress due to advances in 3D Gaussian splatting (3DGS), which provides a favorable trade-off between reconstruction quality and computational efficiency. While previous methods perform well under densely-sampled multi-view settings, acquiring such data in real-world scenarios is often impractical. This limitation has led to growing interest in the sparse-view reconstruction task (Wang et al., 2023a; Yang et al., 2023; Truong et al., 2023; Zhang et al., 2024a; Bao et al., 2025), where only a few input views are available, posing additional challenges for accurate and consistent novel view synthesis.

To address this challenge, existing works (Park et al., 2025) suggest that 3DGS models trained on sparse views tend to overfit a limited set of Gaussian primitives. Therefore, they typically adopt a dropout strategy during training, uniformly dropping Gaussian primitives to reduce over-reconstruction. However, we observe that uniform dropout can inadvertently hurt both well-fitted and under-fitted regions, thereby degrading reconstruction quality in critical areas, as shown in the bottom-right of Figure 1. Moreover, the visualizations of Gaussian distributions in dense- and sparse-view settings (55 and 3 input images) reveal two key issues: over-reconstruction in texture-rich and near-camera regions, leading to dense, aliased Gaussians and rendering artifacts; under-reconstruction in distant areas, where sparse Gaussians fail to capture structural details, resulting in blurry reconstructions.

Based on these observations, we shift our focus to enhancing and evaluating the robustness of 3DGS models under sparse-view settings through both methodological design and evaluation metric. The proposed $D^2GS$ method aims to dynamically adjust the degree of reconstruction based on depth and density information, while the evaluation metric is used to assess the robustness of 3DGS models in a consistent training setting. Specifically, the $D^2GS$ mainly consists of two key components: a **D**epth-and-**D**ensity guided **D**ropout (DD-Drop) mechanism and **D**istance-**A**ware **F**idelity **E**nhancement (DAFE), to improve the stability and spatial completeness of scene reconstruction under sparse-view settings. DD-Drop assigns each Gaussian a dropout score based on local density and camera distance, indicating regions prone to overfitting. High-scoring Gaussians would be dropped with a higher probability to suppress aliasing and improve rendering fidelity. In addition, DAFE avoids underfitting by boosting supervision in distant regions using depth priors.

To further assess the robustness of 3DGS models under sparse-view constraints, we propose a novel evaluation metric, Inter-Model Robustness (IMR), which measures the stability of the learned 3D Gaussian distributions. Specifically, IMR quantifies the consistency of independently trained models by comparing their output Gaussian distributions under identical input settings, reflecting robustness to initialization and training noise. This distribution-based metric complements traditional image-space metrics such as PSNR and SSIM, providing a more direct evaluation of 3D representation quality. We comprehensively evaluate the proposed $D^2GS$ framework on the LLFF and Mip-NeRF360 datasets. Extensive ablation studies further validate the effectiveness of each proposed module. In summary, the main contributions can be summarized as follows:

- We systematically analyze the failure modes of 3DGS in sparse-view settings, revealing consistent patterns of overfitting and underfitting in Gaussian primitives.
- Based on these insights, we propose a unified $D^2GS$ framework that incorporates two complementary modules: a Depth-and-Density Guided Dropout mechanism to suppress overfitting in redundant and dense regions, and a Distance-Aware Fidelity Enhancement module to enhance reconstruction fidelity in underfitting areas.
- To better evaluate the quality of 3D Gaussian representations, we introduce a Gaussian-distribution-based metric to assess robustness and fidelity beyond conventional 2D evaluations. Extensive experiments demonstrate that $D^2GS$ achieves state-of-the-art novel view synthesis while yielding more robust 3D reconstructions.

# 2 RELATED WORK

**Novel View Synthesis.** Novel View Synthesis (NVS) aims to generate unseen views of a scene from given images. Neural Radiance Fields (NeRF) (Mildenhall et al., 2021) reconstruct scenes as

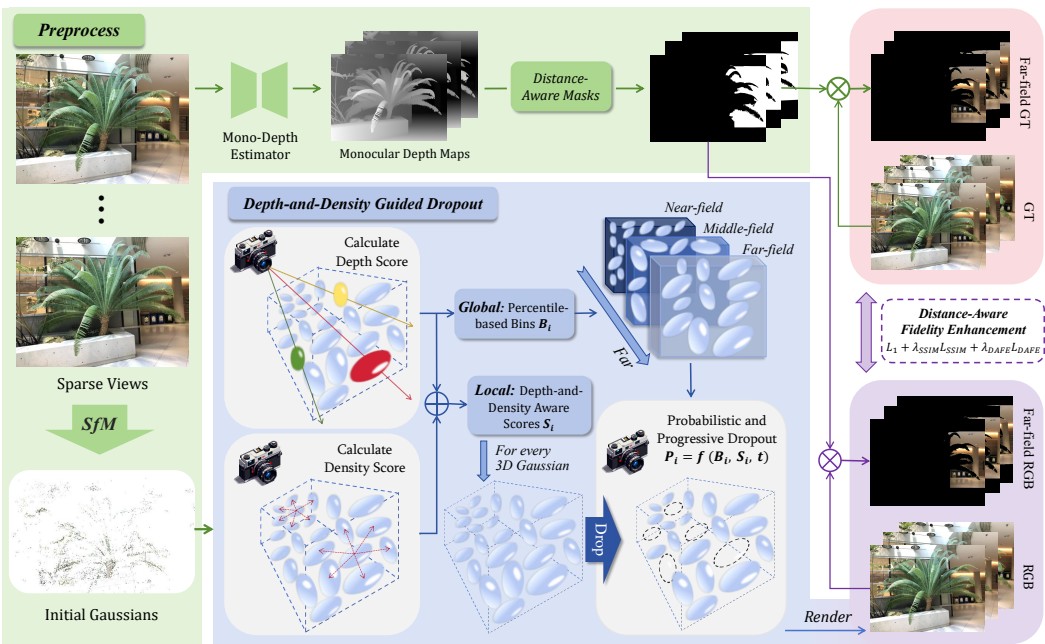

Figure 2: The overall framework of D$^2$GS consists of a Depth-and-Density Guided Dropout (DD-Drop) module and a Distance-Aware Fidelity Enhancement (DAFE) module. The DD-Drop module adaptively removes Gaussian primitives based on depth and density indication through a dual local-global mechanism. The DAFE module enhances supervision for far-field regions using distance-aware masks.

implicit volumetric radiance fields, with many works improving rendering quality (Barron et al., 2021; 2022; Verbin et al., 2022; Chen et al., 2022; Barron et al., 2023) and efficiency (Garbin et al., 2021; Yu et al., 2021; Fridovich-Keil et al., 2022; Müller et al., 2022; Sun et al., 2022; Li et al., 2023; Wang et al., 2023b; Hu et al., 2023). Despite impressive visual fidelity, NeRF-based methods suffer from high computational costs and long training times. To address these limitations, 3D Gaussian Splatting (3DGS) represents scenes with Gaussian primitives and renders via differentiable splatting, achieving real-time synthesis. Building on this, recent methods further enhance reconstruction in diverse 3D vision tasks (Yu et al., 2024; Ye et al., 2024; Lee et al., 2024; Kheradmand et al., 2024; Shi et al., 2025; Niedermayr et al., 2024; Zhang et al., 2024b; Yue et al., 2025).

**Novel View Synthesis with Sparse Views.** NeRF- and 3DGS-based methods have achieved remarkable performance with dense views, but collecting many images is often impractical in real-world scenarios, resulting in significant performance degradation for conventional approaches. To mitigate this, previous NeRF variants introduce architectural enhancements such as semantic consistency (Jain et al., 2021; Qi et al., 2022; 2023), depth supervision (Deng et al., 2022; Niemeyer et al., 2022; Yang et al., 2025; Roessle et al., 2022; Qi et al., 2024; Wang et al., 2023a), frequency regularization (Yang et al., 2023), and cross-view consistency (Truong et al., 2023; Qi et al., 2021). With more efficient 3DGS frameworks, recent methods improve scene understanding via pseudo-view generation (Zhang et al., 2024a), address sparse initialization with additional priors (Bao et al., 2025), and mitigate overfitting to training views (Park et al., 2025; Chen et al., 2025). Recently, some feed-forward methods further advance sparse-view NVS: PixelSplat (Charatan et al., 2024) predicts 3D Gaussian parameters directly from images, MVSplat (Chen et al., 2024) incorporates multi-view stereo cues to improve depth reliability under sparse inputs, and HiSplat (Tang et al., 2024) adopts a hierarchical Gaussian representation to enhance geometric detail and view consistency.

## 3 PROPOSED METHOD

Figure 2 presents the overall pipeline of the proposed D$^2$GS, which takes sparse-view images as input and generates initial point clouds and camera poses through Structure-from-Motion (SfM).

During training, two key modules are introduced: Depth-and-Density Guided Dropout, which regularizes near-field Gaussians via depth- and density-aware dropout; and Distance-Aware Fidelity Enhancement, which strengthens far-field supervision using depth-derived masks predicted by a monocular depth estimator. In the following subsections, we detail the motivation, design, and function of each component, and introduce a dedicated robustness metric for 3DGS under sparse supervision.

## 3.1 MOTIVATION

Our motivation arises from a comprehensive analysis of key factors affecting the performance and stability of sparse-view 3D Gaussian Splatting (3DGS). Figure 1 compares the trained Gaussian primitives under dense- and sparse-view settings. It reveals a significant spatial imbalance: Gaussians are over- and under-fitted in near-and far-field regions.

Specifically, in near-field regions, models trained with only three views (e.g., DropGaussian) produce a much higher density of Gaussians than the dense-view model. In the green box, previous methods generated 11,450 Gaussian primitives, far exceeding the 6,112 Gaussian primitives of the dense view, indicating clear local overfitting. After rendering, we observe that local over-reconstruction in the near field can introduce artifacts that propagate globally, which significantly degrade the rendered image quality. In contrast, far-field regions suffer from underfitting due to limited visibility in training data and frequent occlusion by densely populated near-field Gaussians. In the red box, previous methods generated 3,082 Gaussian primitives, which is noticeably fewer than the 5,224 Gaussian primitives of the dense view, preventing the optimizer from effectively supervising these regions. Therefore, the model is unable to capture accurate geometry and texture in distant areas, leading to blurred or discontinuous structures in the rendered outputs.

## 3.2 DEPTH-AND-DENSITY GUIDED DROPOUT

As observed, near-field regions with high Gaussian density are more susceptible to overfitting. To alleviate this, we propose a spatially adaptive dropout strategy guided by both depth and density. Furthermore, to tackle the problem from both continuous and discrete perspectives, we incorporate two complementary penalty mechanisms operating from local and global viewpoints.

We first introduce the local dropout mechanism, which evaluates the spatial variation of each Gaussian primitive $i = 1, 2, \ldots, N$ based on its depth $d_i$ (Euclidean distance to the camera) and local density $\rho_i$ (estimated via k-nearest neighbors). Both $d_i$ and $\rho_i$ are processed with min–max normalization to obtain the depth score $\tilde{d}_i$ and density score $\tilde{\rho}_i$, respectively. The dropout score $S_i$ is then computed as a weighted combination of the two:

$$S_i = \omega_{depth}\, \tilde{d}_i + \omega_{density}\, \tilde{\rho}_i, \tag{1}$$

where $\omega_{depth}$ and $\omega_{density}$ are weighting coefficients that satisfy $\omega_{\text{depth}} + \omega_{\text{density}} = 1$. This continuous scoring function captures fine-grained local spatial variation, but local information alone is insufficient to characterize overfitting patterns across the entire scene.

The global mechanism is motivated by a depth-induced imbalance: regions at different depth ranges receive markedly different visibility, leading to significantly different overfitting behaviors at a global level. To model this pattern, we further divide the point cloud into three depth-based layers: near, middle, and far. The division is determined by the first and second tertiles of the depth distribution, denoted as thresholds $D_{\text{near}}$ and $D_{\text{middle}}$. Here, our method aims to introduce depth prior information without strongly relying on such partitioning. Each layer is assigned a different attenuation factor, where $\lambda_{\text{middle}}$ and $\lambda_{\text{far}}$ satisfy $0 < \lambda_{\text{far}} < \lambda_{\text{middle}} < 1$, and the near layer uses no attenuation.

This combination of locally continuous and globally discrete mechanisms facilitates fine-grained local tuning while preserving global structural coherence, ultimately leading to efficient control over the overall spatial distribution. This combined design controls the probability of per-Gaussian dropout in a soft and progressive manner, and the corresponding formulation is given by:

$$P_i = \begin{cases} S_i, & d_i \leq D_{near}, \\ \lambda_{middle}\, S_i, & D_{near} < d_i \leq D_{middle}, \\ \lambda_{far}\, S_i, & d_i > D_{middle}, \end{cases} \tag{2}$$

where $P_i$ indicates dropout rate of $i^{th}$ Gaussian primitive. Based on experimental experience, we set $\lambda_{\text{far}} = 0.3$ and $\lambda_{\text{middle}} = 0.7$ in practice.

As the training progresses, the number of Gaussian primitives increases through continuous optimization and refinement. To maintain effective regularization, we gradually increase the dropout ratio over training iterations using a time-dependent global rate $r(t)$, which progressively increases the fraction of Gaussians discarded in later training stages:

$$r(t) = r_{\min} + \left(r_{\max} - r_{\min}\right) \frac{\min(t, T)}{T}, \tag{3}$$

where $t$ denotes the current training step, $r_{\max}$ and $r_{\min}$ are the minimum and maximum dropout rates, and $T$ is the total number of training steps.

### 3.3 DISTANCE-AWARE FIDELITY ENHANCEMENT

To address underfitting in distant regions with missing details, we introduce a Distance-Aware Fidelity Enhancement (DAFE) module that reinforces dedicated supervision in these areas. Specifically, we first employ a monocular depth estimation model to generate depth maps for each input image. These maps are then processed using a depth-thresholding strategy to construct a binary mask that separates the image into near and far regions. The binary distant-region mask $M_{\text{dis}} \in \{0, 1\}^{H \times W}$ is constructed as follows:

$$M_{\text{dis}}(x, y) = \begin{cases} 1, & \text{if } D(x, y) > \tau \, D_{max}, \\ 0, & \text{otherwise}, \end{cases} \tag{4}$$

where $D(x, y)$ is the estimated depth value at pixel $(x, y)$, $D_{max}$ is the maximum depth value, and $\tau$ is the predefined depth threshold.

We then leverage the distant-region mask $M_{\text{dis}}(x, y)$ to modulate the training objective, with the aim of amplifying the supervision signal in under-fitted far-field regions. Specifically, the mask is applied to both the ground-truth image and the rendered output to isolate distant content. A dedicated distance-enhanced loss is computed by measuring their difference in these masked regions:

$$L_{\text{DAFE}} = \frac{1}{\sum M_{\text{dis}}} \sum_{x,y} M_{\text{dis}}(x, y) \cdot \left\| \hat{I}(x, y) - I(x, y) \right\|_1, \tag{5}$$

where $\hat{I}$ and $I$ denote the rendered and ground-truth images respectively. By incorporating $L_{\text{DAFE}}$, the model is guided to allocate greater attention to distant regions during training, which in turn encourages the generation of a denser set of Gaussian primitives in these areas. The improved coverage of Gaussians facilitates more accurate reconstruction of fine-grained details, thereby enhancing the visual quality of novel views in far-field regions.

Following 3D Gaussian splatting, the color reconstruction loss consists of an L1 loss and a D-SSIM loss. Accordingly, the overall training objective is formulated as:

$$L_{\text{total}} = L_1(\hat{I}, I) + \lambda_{\text{SSIM}} L_{\text{D-SSIM}}(\hat{I}, I) + \lambda_{\text{DAFE}} L_{\text{DAFE}}(\hat{I}, I), \tag{6}$$

where $\lambda_{\text{SSIM}}$ and $\lambda_{\text{DAFE}}$ are weighting coefficients that balance the contributions of the D-SSIM and the DAFE loss.

### 3.4 INTER-MODEL ROBUSTNESS ASSESSMENT

As shown in Figure 3 (left), repeated training using the same algorithm and configuration can produce results with considerable variance, leading to large discrepancies in rendering quality. This highlights the importance of quantifying the divergence among independently trained models under identical settings to assess model robustness. To this end, we propose Inter-Model Robustness (IMR), a novel metric specifically designed for 3DGS, grounded in the theory of 2-Wasserstein Distance (Vaserstein, 1969) and Optimal Transport (OT) (Kantorovich, 1960) over Gaussian point clouds, as illustrated in Figure 3 (right).

Let $G_1, G_2, \ldots, G_n$ denote $n$ independently trained 3DGS models, where each model $G_i$ consists of $K_i$ Gaussian primitives:

$$G_i = \{(m_{i,j}, s_{i,j}, q_{i,j}, \alpha_{i,j}, f_{i,j})\}_{j=1}^{K_i}, \tag{7}$$

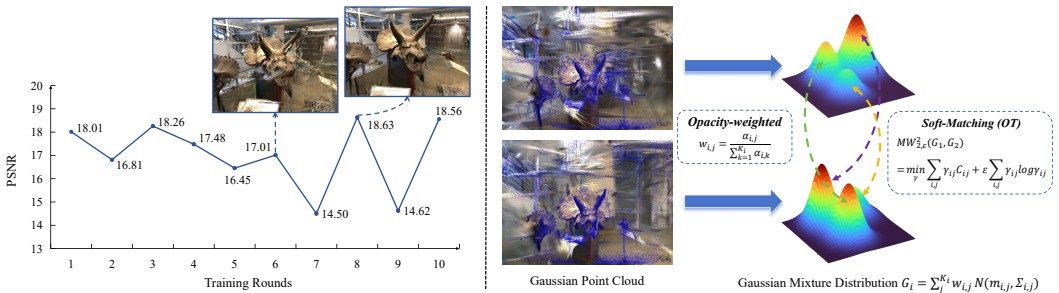

Figure 3: Left: The instability phenomenon of the previous method. PSNR fluctuates significantly across different training rounds, and the quality of the rendered images is highly inconsistent. Right: Calculation procedure of the IMR. The Gaussian point clouds are abstracted as Gaussian mixture distributions, and the 2-Wasserstein Distance and Optimal Transport are used.

where $m_{i,j} \in R^3$ is the center, $s_{i,j} \in R^3$ is the scaling factor, $q_{i,j} \in R^4$ is the rotation, $\alpha_{i,j} \in R$ is the opacity for rendering ,and $f_{i,j} \in R^L$ is an $L$-dimensional color feature. Each Gaussian influences a 3D point $x$ in 3D space following the 3D Gaussian distribution:

$$G_{i,j}(x) = \frac{1}{(2\pi)^{\frac{3}{2}}|\Sigma_{i,j}|^{\frac{1}{2}}} \exp(-\frac{1}{2}(x - m_{i,j})^T \Sigma_{i,j}^{-1}(x - m_{i,j})), \tag{8}$$

where the covariance matrix $\Sigma_{i,j}$ is computed from the scale $s_{i,j}$ and rotation $q_{i,j}$.

To enable robustness analysis, each model is abstracted as a Gaussian mixture distribution:

$$G_i = \sum_{j=1}^{K_i} w_{i,j} \cdot N(m_{i,j}, \Sigma_{i,j}), \quad w_{i,j} = \frac{\alpha_{i,j}}{\sum_{k=1}^{K_i} \alpha_{i,k}}. \tag{9}$$

Here, opacity $\alpha_{i,j}$ serves as a proxy for the importance of each Gaussian in the final rendering, enabling a principled weighting of geometric features during comparison.

For two Gaussian point clouds, it is difficult to directly pair tens of thousands of Gaussian primitives one by one. Therefore, to quantify the difference between two such Gaussian mixtures, we employ the 2-Wasserstein distance and OT theory to establish a soft matching. For two Gaussian distributions $\mu_1 = N(m_1, \Sigma_1)$ and $\mu_2 = N(m_2, \Sigma_2)$, the Wasserstein distance admits a closed-form via the Bures metric (Bures, 1969; Dowson & Landau, 1982):

$$W_2^2(\mu_1, \mu_2) = \|m_1 - m_2\|^2 + \text{tr}(\Sigma_1 + \Sigma_2 - 2(\Sigma_2^{\frac{1}{2}}\Sigma_1\Sigma_2^{\frac{1}{2}})^{\frac{1}{2}}). \tag{10}$$

This expression captures both the positional distance and the shape difference between two ellipsoidal Gaussians. To avoid expensive matrix square roots and improve numerical stability, we approximate the Bures shape term via a first-order Taylor expansion, resulting in following expression:

$$\tilde{W}_2^2(\mu_1, \mu_2) = \|m_1 - m_2\|^2 + \frac{1}{4}\text{tr}\big((\Sigma_1 - \Sigma_2)\Sigma_2^{-1}(\Sigma_1 - \Sigma_2)\big). \tag{11}$$

The detailed mathematical derivation is presented in the Appendix A. Let $G_1$ and $G_2$ denote two 3DGS models. The corresponding mixture Wasserstein distance is then formulated as an OT problem over the Gaussian components (Rubner et al., 2000):

$$\text{MW}_2^2(G_1, G_2) = \min_{\gamma \geq 0} \sum_{i=1}^{K_1} \sum_{j=1}^{K_2} \gamma_{ij} \tilde{W}_2^2(\mu_{1,i}, \mu_{2,j}), \quad \text{s.t.} \sum_j \gamma_{ij} = w_{1,i}, \quad \sum_i \gamma_{ij} = w_{2,j}. \tag{12}$$

This formulation performs soft structure-aware alignment established by the optimal transport plan $\gamma \in R^{K_1 \times K_2}$, eliminating the need for explicit correspondence. To compute the distance at scale, we introduce entropic regularization and solve the relaxed problem using the Sinkhorn algorithm (Sinkhorn & Knopp, 1967; Cuturi, 2013):

$$\text{MW}_{2,\varepsilon}^2(G_1, G_2) = \min_{\gamma} \sum_{i,j} \gamma_{ij} C_{ij} + \varepsilon \sum_{i,j} \gamma_{ij} \log \gamma_{ij}, \tag{13}$$

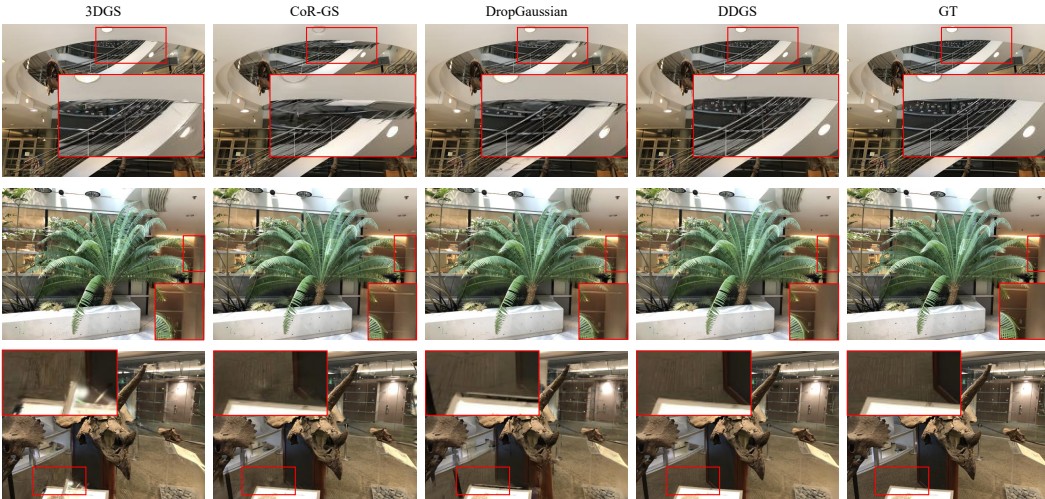

Figure 4: Qualitative Comparison on LLFF dataset (Mildenhall et al., 2019). Comparisons were conducted with 3DGS, CoR-GS, DropGaussian. Our method effectively avoids the artifacts and maintains accurate reconstructions.

| | Methods | LLFF (3-view 1/8 Resolution) | | | | LLFF (3-view 1/4 Resolution) | | | |
|---|---|---|---|---|---|---|---|---|---|
| | | PSNR(↑) | SSIM(↑) | LPIPS (↓) | AVGE(↓) | PSNR(↑) | SSIM(↑) | LPIPS (↓) | AVGE(↓) |
| NeRF-based | Mip-NeRF (Barron et al., 2021) | 16.11 | 0.401 | 0.460 | 0.206 | 15.22 | 0.351 | 0.540 | 0.236 |
| | DietNeRF (Jain et al., 2021) | 14.94 | 0.370 | 0.496 | 0.233 | 13.86 | 0.305 | 0.578 | 0.271 |
| | RegNeRF (Niemeyer et al., 2022) | 19.08 | 0.587 | 0.336 | 0.139 | 18.66 | 0.535 | 0.411 | 0.156 |
| | FreeNeRF (Yang et al., 2023) | 19.63 | 0.612 | 0.308 | 0.128 | 19.13 | 0.562 | 0.384 | 0.146 |
| | SparseNeRF (Wang et al., 2023a) | 19.86 | 0.624 | 0.328 | 0.128 | 19.07 | 0.564 | 0.392 | 0.147 |
| 3DGS-based | 3DGS (Kerbl et al., 2023) | 19.22 | 0.649 | 0.229 | 0.118 | 16.94 | 0.488 | 0.402 | 0.180 |
| | DNGaussian (Li et al., 2024) | 19.12 | 0.591 | 0.294 | 0.132 | 18.47 | 0.578 | 0.330 | 0.145 |
| | FSGS (Zhu et al., 2024) | 20.43 | 0.682 | 0.248 | 0.108 | 19.71 | 0.642 | 0.283 | 0.122 |
| | CoR-GS (Zhang et al., 2024a) | 20.45 | 0.712 | 0.196 | 0.092 | 19.96 | 0.696 | 0.250 | 0.119 |
| | LoopSparseGS (Bao et al., 2025) | 20.85 | 0.717 | 0.205 | 0.096 | 20.19 | 0.680 | 0.274 | 0.114 |
| | DropGaussian (Park et al., 2025) | 20.76 | 0.713 | 0.200 | 0.097 | 20.01 | 0.690 | 0.258 | 0.113 |
| | D²GS (Ours) | 21.35 | 0.746 | 0.179 | 0.087 | 20.56 | 0.695 | 0.254 | 0.107 |

Table 1: Performance comparisons of sparse-view synthesis on LLFF dataset. The best, second-best, and third-best entries are marked in red, orange, and yellow, respectively.

where $C_{ij} = \tilde{W}_2^2(N(m_{1,i}, \Sigma_{1,i}), N(m_{2,j}, \Sigma_{2,j}))$ is the cost matrix, and $\varepsilon > 0$ is the regularization strength. With the introduction of entropic regularization to the original discrete optimal transport objective, the mixture Wasserstein distance between 3DGS models admits a unique and well-defined optimal solution (Delon & Desolneux, 2020).

Direct computation of transport between tens of thousands of Gaussians is computationally infeasible. To further improve tractability, we adopt a depth-stratified importance sampling strategy to select approximately 10,000 Gaussians primitives. Given that far-field Gaussians are more prone to noise and instability due to underfitting, they are oversampled accordingly.

Let $S_{ij} = \text{MW}_2^2(G_i, G_j)$ denote the pairwise distances between $N$ independently trained models. To specifically penalize model pairs with large divergence, we use a weighted formulation that amplifies the impact of inconsistent models. Finally, we define the Inter-model Robustness (IMR) metric as:

$$\text{IMR} = ln\left(\frac{\sum_{1 \le i < j \le N} S_{ij}^2}{\sum_{1 \le i < j \le N} S_{ij}}\right) \tag{14}$$

| Methods | PSNR(↑) | SSIM(↑) | LPIPS(↓) | AVGE(↓) |
|---|---|---|---|---|
| 3DGS | 18.52 | 0.523 | 0.415 | 0.159 |
| FSGS | 18.80 | 0.531 | 0.418 | 0.156 |
| CoR-GS | 19.52 | 0.558 | 0.418 | 0.146 |
| DropGaussian | 19.74 | 0.577 | 0.364 | 0.136 |
| D$^2$GS (Ours) | 20.09 | 0.587 | 0.356 | 0.130 |

Table 2: Performance comparisons of sparse-view synthesis on MipNeRF360 dataset. The best, second-best, and third-best entries are marked in red, orange, and yellow, respectively.

## 4 EXPERIMENTS

We conduct experiments on LLFF (Mildenhall et al., 2019) and Mip-NeRF360 (Barron et al., 2022), following the same data splits and downsampling as prior work. Our implementation is built on DropGaussian, with 10k training iterations per dataset. The evaluation process uses PSNR, SSIM, LPIPS, and AVGE (the geometric mean of MSE $= 10^{-\frac{\text{PSNR}}{10}}$, $\sqrt{1 - \text{SSIM}}$, LPIPS), along with our proposed IMR for robustness. All experiments run on a single H20 GPU. More Implementation Details are presented in the Appendix B.

**Quantitative evaluation.** We compare D$^2$GS with some NeRF-based methods (Mip-NeRF, Diet-NeRF, RegNeRF, FreeNeRF, SparseNeRF) and 3DGS-based methods (3DGS, DNGaussian, FSGS, CoR-GS, LoopSparseGS, DropGaussian) on LLFF and MipNeRF360. As shown in Tables 1 and 2, D$^2$GS consistently achieves the best results. On LLFF (1/8 res.), D$^2$GS surpasses FSGS, CoR-GS, and LoopSparseGS by 0.92/0.9/0.5 dB PSNR with notable SSIM/LPIPS/AVGE gains, and outperforms DropGaussian by 0.59/0.55 dB at 1/8 and 1/4 res. On MipNeRF360, it also improves over CoR-GS and DropGaussian by 0.57 dB and 0.35 dB PSNR, respectively, confirming its superior reconstruction quality. These gains largely stem from the proposed DD-Drop and DAFE modules, which jointly suppress overfitting in near-field regions while enhancing distant details. More results are presented in the Appendix E.

To assess the robustness of the trained 3D Gaussian primitives, we report the metric IMR, measuring the dispersion across independently trained models. The number of Gaussian primitives in the scenes of LLFF ranges from 20k to 310k. Table 3 shows that our method achieves the lowest IMR in both sparse settings: 3.039 (3-view) and 3.109 (6-view), respectively. This indicates more stable and consistent Gaussian reconstructions across runs.

| Methods | IMR(↓) | |
|---|---|---|
| | LLFF (3-view) | LLFF (6-view) |
| 3DGS | 3.162 | 3.234 |
| CoR-GS | 3.136 | 3.270 |
| DropGaussian | 3.205 | 3.143 |
| D$^2$GS (Ours) | 3.039 | 3.109 |

Table 3: IMR comparison on LLFF Dataset with 3-view and 6-view Settings. All results are tested on ten independent training models.

**Qualitative evaluation.** Figure 4 shows qualitative results on LLFF, comparing 3DGS, CoR-GS, DropGaussian, D$^2$GS, and GT. As highlighted by the red boxes, D$^2$GS yields sharper details and fewer artifacts, preserving more high-frequency structures than DropGaussian with a random dropout strategy. This visual comparison highlights the superiority of D$^2$GS in reconstructing fine-grained geometry under sparse views. These improvements come mainly from the targeted suppression of redundant Gaussians by DD-Drop module and the enhancement of distant structures.

**Ablation study on the proposed components.** We conduct ablation experiments to validate the effectiveness of each proposed modules on LLFF, as summarized in Table 4. Starting from the baseline without any proposed component, we progressively add the density score, depth score, and depth-based layering for DD-Drop, each of which steadily improves PSNR, SSIM, LPIPS, and IMR. Finally, incorporating the DAFE module further enhances reconstruction quality, leading to the best overall performance. These results confirm that all components contribute complementary benefits, with the full model achieving the highest visual fidelity.

**Ablation study on DD-Drop.** The upper left part of Table 5 shows different weights $\omega_{depth}$ and $\omega_{density}$ to balance the influence of normalized depth and density scores in the dropout process. The best performance is achieved when $\omega_{depth} = 0.5$ and $\omega_{density} = 0.5$, suggesting that both depth and density contribute positively, and overly increasing the weight of either factor results in

| Density Score | Depth Score | Depth-based Layering | DAFE | PSNR(↑) | SSIM(↑) | LPIPS(↓) | IMR(↓) |
|---|---|---|---|---|---|---|---|
| | | | | 19.22 | 0.649 | 0.229 | 3.162 |
| ✓ | | ✓ | | 21.02 | 0.732 | 0.191 | 3.119 |
| | ✓ | ✓ | | 20.92 | 0.728 | 0.200 | 3.155 |
| ✓ | ✓ | | | 21.10 | 0.735 | 0.187 | 3.111 |
| ✓ | ✓ | ✓ | | 21.17 | 0.740 | 0.181 | 3.088 |
| ✓ | ✓ | ✓ | ✓ | **21.35** | **0.746** | **0.179** | **3.039** |

Table 4: Ablation Study on proposed components. The ✓ indicates adding the module.

| $r_{\min}$ | $r_{\max}$ | PSNR(↑) | SSIM(↑) | LPIPS(↓) | $\tau$ (%) | PSNR(↑) | SSIM(↑) | LPIPS(↓) |
|---|---|---|---|---|---|---|---|---|
| 0.05 | 0.3 | 21.16 | 0.740 | 0.181 | 5 | 21.25 | 0.744 | 0.180 |
| 0.1 | 0.3 | 21.11 | 0.740 | 0.181 | 10 | 21.26 | 0.743 | 0.180 |
| 0.05 | 0.5 | 21.06 | 0.738 | 0.187 | 15 | 21.20 | 0.741 | 0.181 |
| $\omega_{depth}$ | $\omega_{density}$ | PSNR(↑) | SSIM(↑) | LPIPS(↓) | $\lambda_{\text{DAFE}}$ | PSNR(↑) | SSIM(↑) | LPIPS(↓) |
| 0.2 | 0.8 | 21.07 | 0.737 | 0.183 | 0.5 | 21.27 | 0.743 | 0.180 |
| 0.5 | 0.5 | 21.16 | 0.740 | 0.181 | 1.0 | 21.30 | 0.744 | 0.179 |
| 0.8 | 0.2 | 21.04 | 0.734 | 0.190 | 1.5 | 21.25 | 0.743 | 0.182 |

Table 5: Ablation study on different parameters in our model. In DD-Drop module, $r_{\min}$ and $r_{\max}$ denote the minimum and maximum dropout rates, while $\omega_{depth}$ and $\omega_{density}$ are used in computing the dropout score. In DAFE module, $\tau$ denotes the depth threshold controlling the proportion of far regions retained, and $\lambda_{\text{DAFE}}$ denotes the weight of the DAFE loss.

a performance drop. The lower left part of Table 5 presents results under different combinations of minimum and maximum dropout thresholds $r_{\min}$ and $r_{\max}$, which control the dynamic range of the time-dependent dropout rate $r(t)$. We observe that setting $r_{\min} = 0.05$ and $r_{\max} = 0.3$ achieves the best performance, indicating that maintaining a mild dropout rate in the early training stages helps to preserve the geometry of the essential scene, while gradually increasing the dropout rate to a moderate level encourages effective regularization during the later stages.

**Ablation study on DAFE.** We further conduct ablations on the components of the proposed DAFE loss. As shown in the upper right part of Table 5, we compare different values for the depth-based masking ratio, where selecting the top 5% of the farthest depth values yields the best performance, indicating that enforcing depth fidelity in distant regions is particularly beneficial under sparse-view settings. In the lower right part of Table 5 investigates the impact of the weighting hyperparameter in the DAFE loss. A moderate value (e.g., 1.0) provides the best trade-off across all metrics.

Table 6 compares different depth estimation models on DAFE supervision. DepthAnything V2 is used by default. While different depth estimator has an impact on the performance, our method demonstrates consistent improvements across all models, indicating that DAFE is compatible with a variety of depth priors and can effectively enhance rendering quality under sparse-view settings.

| Methods | PSNR(↑) | SSIM(↑) | LPIPS(↓) |
|---|---|---|---|
| MiDaS | 21.21 | 0.740 | 0.182 |
| DPT | 21.27 | 0.743 | 0.181 |
| DepthAnything V2 | **21.35** | **0.746** | **0.179** |

Table 6: Ablation Study on different monocular depth estimators: MiDas (Ranftl et al., 2022) with VIT-small backbone, DPT (Ranftl et al., 2021) with VIT-Hybrid backbone, and DepthAnything V2 (Yang et al., 2024).

## 5  CONCLUSION

In this work, we present a novel $D^2GS$ for enhancing sparse-view 3D reconstruction. We introduce a depth-and-density guided dropout that selectively removes over-fitted Gaussians in texture-dense, near-camera regions. To complement this, the proposed Distance-Aware Fidelity Enhancement loss leverages depth priors to reinforce geometric consistency—particularly in distant regions prone to underfitting. Beyond accuracy, we also assess robustness with an inter-model robustness metric, showing more stable Gaussian distributions across runs. Extensive experiments on standard benchmarks confirm consistent gains in both quantitative metrics and visual fidelity over strong baselines.

**Acknowledgements.** This work has been supported by the New Cornerstone Science Foundation through the XPLORER PRIZE.

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

# A  BURES DISTANCE APPROXIMATION

Let $G_1, G_2, \ldots, G_n$ denote $n$ independently trained 3DGS models, where each model $G_i$ consists of $K_i$ Gaussian primitives:

$$G_i = \{(m_{i,j}, s_{i,j}, q_{i,j}, \alpha_{i,j}, f_{i,j})\}_{j=1}^{K_i}, \tag{15}$$

where $m_{i,j} \in R^3$ is the center, $s_{i,j} \in R^3$ is the scaling factor, $q_{i,j} \in R^4$ is the rotation, $\alpha_{i,j} \in R$ is the opacity for rendering ,and $f_{i,j} \in R^L$ is an $L$-dimensional color feature. Each Gaussian influences a 3D point $x$ in 3D space following the 3D Gaussian distribution:

$$G_{i,j}(x) = \frac{1}{(2\pi)^{\frac{3}{2}}|\Sigma_{i,j}|^{\frac{1}{2}}} \exp(-\frac{1}{2}(x - m_{i,j})^T \Sigma_{i,j}^{-1}(x - m_{i,j})), \tag{16}$$

where the covariance matrix $\Sigma_{i,j}$ is computed from the scale $s_{i,j}$ and rotation $q_{i,j}$. To enable robustness analysis, each model is abstracted as a Gaussian mixture distribution:

$$G_i = \sum_{j=1}^{K_i} w_{i,j} \cdot N(m_{i,j}, \Sigma_{i,j}), \quad w_{i,j} = \frac{\alpha_{i,j}}{\sum_{k=1}^{K_i} \alpha_{i,k}}. \tag{17}$$

Here, opacity $\alpha_{i,j}$ serves as a proxy for the importance of each Gaussian in the final rendering, enabling a principled weighting of geometric features during comparison.

To quantify the difference between two such Gaussian mixtures, we employ 2-Wasserstein distance. For two Gaussian distributions $\mu_1 = N(m_1, \Sigma_1)$ and $\mu_2 = N(m_2, \Sigma_2)$, the Wasserstein distance admits a closed-form via the Bures metric:

$$W_2^2(\mu_1, \mu_2) = \|m_1 - m_2\|^2 + \text{tr}(\Sigma_1 + \Sigma_2 - 2(\Sigma_2^{\frac{1}{2}} \Sigma_1 \Sigma_2^{\frac{1}{2}})^{\frac{1}{2}}). \tag{18}$$

This expression captures both the positional distance and the shape difference between two ellipsoidal Gaussians.

To avoid the computational cost and potential numerical instability associated with computing matrix square roots in the Bures distance, we adopt a first-order Taylor approximation for the shape-related term. Specifically, we focus on approximating the trace expression in the closed-form formula for the squared 2-Wasserstein distance. Define the shape term as:

$$\Phi(\Sigma_1, \Sigma_2) = \text{tr}\left(\Sigma_1 + \Sigma_2 - 2\left(\Sigma_2^{1/2} \Sigma_1 \Sigma_2^{1/2}\right)^{1/2}\right). \tag{19}$$

To simplify this expression, we perform a change of basis by whitening $\Sigma_2$, such that the resulting expression is centered around the identity matrix. This transforms the matrix product into a form suitable for series expansion:

$$\Sigma_2^{-1/2} \Sigma_1 \Sigma_2^{-1/2} = I + \Lambda, \quad \text{where } \Lambda = \Sigma_2^{-1/2} \Delta \Sigma_2^{-1/2}, \quad \text{and } \Delta = \Sigma_1 - \Sigma_2. \tag{20}$$

By construction, $\Lambda$ is a symmetric matrix quantifying the normalized deviation between $\Sigma_1$ and $\Sigma_2$, and its Frobenius norm scales with the norm of $\Delta$: Note that $\|\Lambda\|_F = \mathcal{O}(\|\Delta\|_F)$.

We now expand the matrix square root using a Taylor series about the identity:

$$(I + \Lambda)^{1/2} = I + \tfrac{1}{2}\Lambda - \tfrac{1}{8}\Lambda^2 + \mathcal{O}(\|\Lambda\|_F^3), \tag{21}$$

Substituting this back, we obtain the following approximation:

$$\left(\Sigma_2^{1/2} \Sigma_1 \Sigma_2^{1/2}\right)^{1/2} = \Sigma_2^{1/2}(I + \Lambda)^{1/2} \Sigma_2^{1/2}$$
$$= \Sigma_2 + \tfrac{1}{2}\Delta - \tfrac{1}{8}\Delta \Sigma_2^{-1} \Delta + \mathcal{O}(\|\Delta\|_F^3), \tag{22}$$

where the approximation holds up to third-order terms in $|\Delta|_F$. We now substitute the expansion into the definition of the sharp term $\Phi$:

$$\Phi = \text{tr}\left(\Sigma_1 + \Sigma_2 - 2\Sigma_2 - \Delta + \tfrac{1}{4}\Delta \Sigma_2^{-1} \Delta + \mathcal{O}(\|\Delta\|_F^3)\right)$$
$$= \tfrac{1}{4} \text{tr}\left(\Delta \Sigma_2^{-1} \Delta\right) + \mathcal{O}(\|\Delta\|_F^3) \tag{23}$$

| Method | Training Time (s) |
|--------|-------------------|
| FSGS | 425 |
| CoR-GS | 223 |
| DropGaussian | 56 |
| $D^2GS$ (Ours) | 82 |

Table 7: Training time comparison on LLFF Dataset (Mildenhall et al., 2019) with 3-view.

Finally, combining the approximated shape term with the mean term from the original 2-Wasserstein distance, we arrive at the following closed-form approximation:

$$\tilde{W}_2^2(\mu_1, \mu_2) = \|m_1 - m_2\|^2 + \frac{1}{4} \, \text{tr}\big((\Sigma_1 - \Sigma_2)\Sigma_2^{-1}(\Sigma_1 - \Sigma_2)\big) \tag{24}$$

which avoids expensive matrix square roots while preserving second-order accuracy in $\Delta$. This approximation significantly improves efficiency in large-scale scenarios with a huge amount of large-scale 3D Gaussian primitives.

## B  IMPLEMENTATION DETAILS

Following the setup in prior works (Kerbl et al., 2023; Zhu et al., 2024; Park et al., 2025), we begin our pipeline with unstructured multi-view images, which are calibrated using a Structure-from-Motion (SfM). In particular, we employ COLMAP to generate an initial point cloud via dense stereo matching using the "patch-match-stereo" algorithm, followed by stereo fusion to produce a unified point cloud. Next, we initialize the spherical harmonic (SH) coefficients to degree 0. The 3D Gaussian positions are initialized using the fused point cloud, and we set the opacity to 0.1. Other attributes, such as scale and rotation are initially set to zero.

During training, we initialize the spherical harmonics (SH) at degree 0 to provide a coarse approximation of scene lighting. The SH degree is then incrementally increased by 1 every 1,000 iterations, up to a maximum of degree 3. This progressive refinement allows the lighting representation to capture more detailed effects as training proceeds. The learning rates for different parameters are set as follows: 0.00016 for Gaussian position, 0.0025 for SH coefficients, 0.05 for opacity, 0.005 for scale, and 0.001 for rotation. To maintain stable opacity evolution, we reset the opacity values of all Gaussians to 0.01 every 3,000 iterations. Depth information is obtained per iteration, while density information is recalculated every 500 iterations, with $k$ set to 6 in the k-nearest neighbors algorithm. The depth information is defined relative to a randomly selected training camera, consistent with the original 3DGS training setup. As shown in Table 7, the inclusion of these computations increases the training time, but this increase is relatively small and controllable.

## C  DISCUSSION OF DROPOUT

In DropGaussian, an ablation study compares random dropout with selective dropout strategies (e.g., based on gradient or distance), and concludes that random schemes perform better. However, we consider that this observation is not a fundamental limitation of selective dropout itself, but rather a consequence of the hard dropout strategy used in their experiments. Specifically, the selective methods in DropGaussian aggressively remove the "top-k" primitives with the highest scores (e.g., most distant), which introduces several issues:

- **Persistent suppression of specific regions (Discussed in DropGaussian)**: Since the same subset of Gaussians is repeatedly identified as high-score targets, these regions are systematically removed throughout training, causing spatial bias and under-coverage.
- **Over-suppression in detail-rich areas**: In textured or structurally complex regions, high-density Gaussians naturally accumulate. Hard selection blindly eliminates these, inadvertently erasing important details and shifting the model from overfitting to underfitting.

In contrast, our Depth-and-Density Guided Dropout (DD-Drop) introduces a soft, probabilistic selection mechanism. Instead of deterministically removing the most over-represented Gaussians, we

| Methods | LLFF (6-view) | | | | MipNeRF360 (24-view) | | | |
|---|---|---|---|---|---|---|---|---|
| | PSNR(↑) | SSIM(↑) | LPIPS (↓) | AVGE(↓) | PSNR(↑) | SSIM(↑) | LPIPS (↓) | AVGE(↓) |
| 3DGS | 23.80 | 0.814 | 0.125 | 0.061 | 22.80 | 0.708 | 0.276 | 0.092 |
| FSGS | 24.09 | 0.823 | 0.145 | 0.062 | 23.70 | 0.745 | 0.230 | 0.079 |
| CoR-GS | 24.49 | 0.837 | 0.115 | 0.055 | 23.39 | 0.727 | 0.271 | 0.087 |
| DropGaussian | 24.43 | 0.829 | 0.127 | 0.057 | 23.75 | 0.756 | 0.227 | 0.078 |
| $D^2$GS (Ours) | **24.84** | **0.834** | **0.122** | **0.055** | **24.13** | **0.763** | **0.221** | **0.075** |

Table 8: Performance comparisons of sparse-view synthesis on LLFF dataset (Mildenhall et al., 2019) with 6-view and MipNeRF360 dataset (Barron et al., 2022) with 24-view.

| Methods | 3-views | | | | 6-views | | | |
|---|---|---|---|---|---|---|---|---|
| | PSNR(↑) | SSIM(↑) | LPIPS (↓) | AVGE(↓) | PSNR(↑) | SSIM(↑) | LPIPS (↓) | AVGE(↓) |
| DietNeRF | 11.85 | 0.633 | 0.314 | 0.232 | 20.63 | 0.778 | 0.201 | 0.094 |
| RegNeRF | 18.89 | 0.745 | 0.190 | 0.107 | 22.20 | 0.841 | 0.117 | 0.066 |
| FreeNeRF | 19.92 | 0.787 | 0.182 | 0.095 | 23.25 | 0.844 | 0.131 | 0.063 |
| 3DGS | 17.65 | 0.816 | 0.146 | 0.102 | 20.33 | 0.776 | 0.223 | 0.099 |
| FSGS | 17.34 | 0.818 | 0.169 | 0.110 | 21.55 | 0.880 | 0.127 | 0.068 |
| DNGaussian | 18.91 | 0.790 | 0.176 | 0.101 | 22.10 | 0.851 | 0.148 | 0.071 |
| CoR-GS | 19.21 | 0.853 | 0.119 | 0.082 | 24.51 | 0.917 | 0.068 | 0.041 |
| DropGaussian | 20.29 | 0.863 | 0.122 | 0.075 | 24.57 | 0.919 | 0.072 | 0.042 |
| $D^2$GS (Ours) | **21.25** | **0.865** | **0.121** | **0.069** | **25.25** | **0.920** | **0.071** | **0.039** |

Table 9: Performance comparisons of sparse-view synthesis on DTU dataset (Jensen et al., 2014) with 3-view and 6-view.

compute a depth- and density-based dropout score for each primitive and sample dropout actions according to this score distribution. This probabilistic characteristic avoids repeatedly discarding the same Gaussians, encourages diversity in regularization, and better preserves important scene structures. Moreover, our dropout probability is further modulated by a global depth-based layering strategy, which adaptively attenuates dropout strength in middle- and far-field regions to prevent excessive loss of sparsely visible geometry.

This design distinction highlights that the key to effective selective dropout lies not just in what signals are used (e.g., depth or distance), but how they are applied. By combining structured guidance with probabilistic flexibility, DD-Drop avoids the pitfalls of rigid selection and achieves superior performance (as shown in Table 1 to 3 in the main paper), reconciling the strengths of selective and random dropout schemes.

## D  LIMITATIONS

Despite the strong performance of our $D^2$GS framework in mitigating overfitting and underfitting under sparse-view settings, there remain promising directions for further improvement. For instance, while our Depth-and-Density Guided Dropout effectively regularizes the spatial distribution of Gaussians, it relies on hand-crafted depth thresholds and fixed weight coefficients, which may not fully capture complex scene-specific priors. Additionally, our robustness metric IMR focuses on inter-model consistency but does not yet consider perceptual stability under dynamic view synthesis. Exploring adaptive dropout schedules, learnable supervision masks, and temporally-aware robustness metrics presents fruitful avenues for extending our work.

## E  QUANTITATIVE EVALUATION

**Quantitative evaluation.** Due to the unconditional dropout strategy used in DropGaussian, its training exhibits significant instability—especially under settings with more input views. As a result, we found it difficult to reproduce the results reported in their paper, and thus, we report the results obtained from our training. Table 8 reports the quantitative comparisons on the LLFF dataset (6-view) and MipNeRF360 dataset (24-view), evaluated using PSNR, SSIM, LPIPS, and AVGE met-

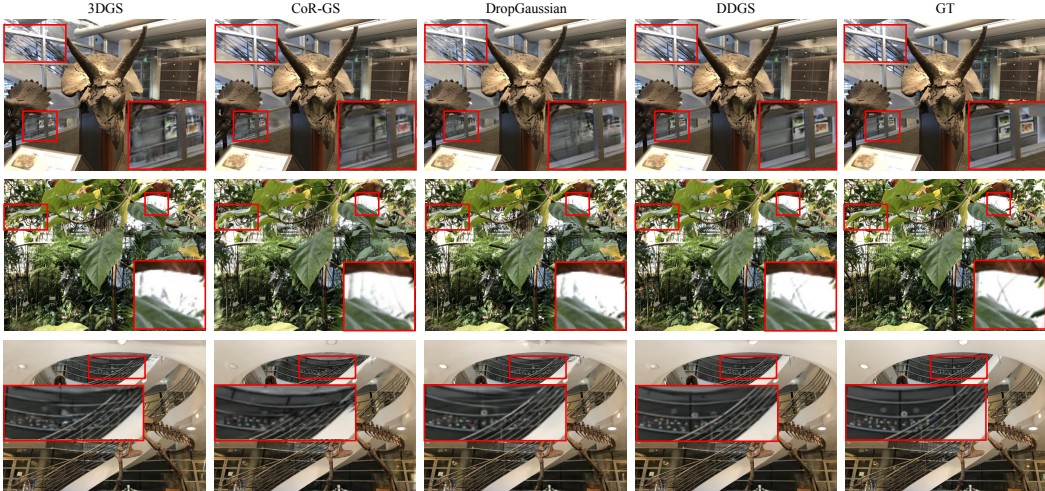

Figure 5: Qualitative Comparison on LLFF dataset (Mildenhall et al., 2019) with 6-view. Comparisons were conducted with 3DGS, CoR-GS, DropGaussian. Our method effectively avoids the artifacts and maintains accurate reconstructions.

rics. Among all methods, our proposed $D^2GS$ consistently achieves the best or highly competitive performance across both datasets and all evaluation metrics.

On the LLFF dataset, $D^2GS$ attains the highest PSNR of 24.84 and the lowest AVGE of 0.055, indicating superior image fidelity and robust 3D reconstruction stability. It also achieves a strong SSIM of 0.834 and a low LPIPS of 0.122, reflecting better structural and perceptual consistency compared to baselines. Similarly, on the MipNeRF360 dataset, $D^2GS$ again outperforms other methods with the best PSNR (24.13), SSIM (0.763), LPIPS (0.221), and AVGE (0.075). Notably, DropGaussian, while competitive, suffers from performance instability due to its unconditional dropout policy, especially under higher input-view scenarios like 24-view. Therefore, we report its results based on our own reproducible implementation.

We have additionally evaluated $D^2GS$ on the DTU dataset (Jensen et al., 2014), which is object-centric and has quite different geometry and depth characteristics compared to LLFF and MipNeRF360. As shown in Table 9, $D^2GS$ consistently outperforms all baseline methods across both 3-view and 6-view settings on all evaluation metrics.

These results demonstrate the effectiveness and robustness of $D^2GS$ under sparse-view constraints across diverse scenes and datasets.

**Qualitative evaluation.** Figure 5 and Figure 6 illustrates qualitative comparisons on the LLFF and MipNeRF360 dataset among 3DGS, CoR-GS, DropGaussian, $D^2GS$. Our method consistently delivers strong performance with more input views, highlighting its robust generalization capability. As emphasized by the red boxes, $D^2GS$ demonstrates sharper details and significantly reduced artifacts, retaining more high-frequency structural information.

## F  LLMs in Paper Writing

Large language models (LLMs, e.g., GPT-4, GPT-5) were used only for refining grammar and sentence structure, with the sole purpose of enhancing readability, clarity, and fluency. They did not contribute to the research ideas, methods, results, or interpretations. All scientific and technical content of this work was conceived, conducted, and written entirely by human authors.

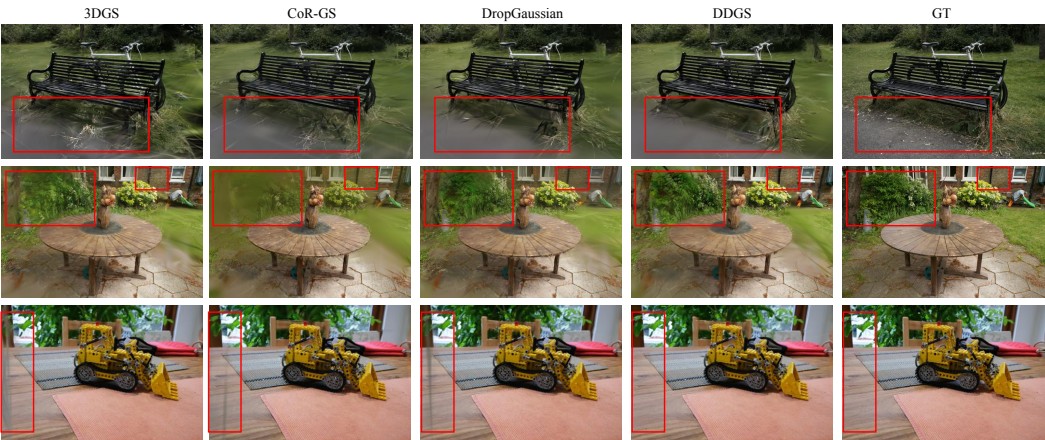

Figure 6: Qualitative Comparison on MipNeRF360 dataset (Barron et al., 2022) with 12-view. Comparisons were conducted with 3DGS, CoR-GS, DropGaussian. Our method effectively avoids the artifacts and maintains accurate reconstructions.

