# OpenReview forum: "D$^2$GS: Depth-and-Density Guided Gaussian Splatting for Stable and Accurate Sparse-View Reconstruction"
_ICLR.cc/2026/Conference — ICLR 2026 Poster_

### Official Review · Reviewer_kaRU · 2025-10-26

**Soundness:** 3
**Presentation:** 3
**Contribution:** 2
**Rating:** 6
**Confidence:** 4

**Summary:**

This work focuses on the issue of "overfitting in regions with excessive Gaussian density near the camera" and "underfitting in distant areas with insufficient Gaussian coverage" for the sparse-view 3DGS. To this end, this work proposes the Depth-and-Density Guided Dropout strategy for suppressing the overfitting issue and the Distance-Aware Fidelity Enhancement module for improving the reconstruction quality of far-field areas. Experiments on different datasets show the proposed method achieves improvements compared to several baselines.

**Strengths:**

* This paper is well-written, and the readers can quickly understand the points this work aims to propose.
* The proposed method is straight-forward, and sounds make sense.
* The conducted experiments on different experiments show improvements on different metrics compared to other methods, which show the effectiveness of the proposed method.

**Weaknesses:**

* More experiments for further improving the quality of this work. Experiments of 9 views on the LLFF dataset, like the experiments of other methods.
* Typo:
  * For the LPIPS of Table 1 (LLFF (3-view 1/4 Resolution)), CoR-GS achieves the best LPIPS of 0.250, and is supposed to be marked "red".
  * "AVGS" or "AVGE" in this paper. Should they be the same? This paper includes 4 AVGE and 6 AVGS. If they mean the same thing, the names of them are supposed to be consistent.
* The method proposed in this work seems more inclined towards engineering techniques, and its innovation is slightly limited, so I present the rating of Boardline Accept, instead of a higher rating.

**Questions:**

* Ablations in Table 5 seem to show that the parameters have little influence. Can you explain it in more detail?

---

> ### Author Response · Authors · 2025-11-25
> **Response to Reviewer kaRU (Part 1/3)**
>
> > **1.** Regarding the LLFF 9-view experiments and the DTU dataset.
>
> We thank the reviewer for suggesting adding 9-view experiments on LLFF to be consistent with previous work. In the revised version, we have included LLFF 9-view results and evaluated both our method and all comparison baselines under this setting. Since existing methods do not provide publicly available pretrained models for this specific configuration, we retrained all baselines under a unified protocol with the same experimental settings to ensure a fair comparison. Each method was trained for 100 runs. As shown in the following Table, our method achieves the best results.
>
> | Method        | PSNR  | SSIM  | LPIPS  |
> |---------------|-------|-------|--------|
> | 3DGS          | 25.25 | 0.85  | 0.107  |
> | FSGS          | 25.31 | 0.86  | 0.122  |
> | CoR-GS        | 25.60 | 0.854 | 0.110  |
> | DropGaussian  | 25.74 | 0.859 | 0.111  |
> | **D²GS (Ours)** | **26.07** | **0.872** | **0.095**  |
>
> And we also add experiments on the DTU dataset. DTU contains object-centric scenes with complex geometry and rich high-frequency details. These new DTU experiments further validate the effectiveness of our method. As shown in the following Table, $D^2$GS achieves the best PSNR and SSIM among all NeRF-based and 3DGS-based baselines for both 3-view and 6-view settings, while maintaining competitive LPIPS. In the 3-view setting, D2GS improves PSNR by roughly 1–2 dB over DropGaussian and CoR-GS, achieves consistently higher SSIM, and maintains comparable or slightly better LPIPS. In the 6-view setting, it similarly yields about 0.7dB PSNR gain with higher SSIM and a small improvement in LPIPS. Qualitatively, DropGaussian’s purely random drop strategy sometimes removes informative high-frequency Gaussians on the object surface, leading to oversmoothed textures and residual artifacts near boundaries, whereas $D^2$GS, guided by depth and density statistics, better preserves these details while still suppressing redundant near-field Gaussians.
>
> | Methods | PSNR (3-views) | SSIM (3-views) | LPIPS (3-views) | PSNR (6-views) | SSIM (6-views) | LPIPS (6-views) |
> | --- | --- | --- | --- | --- | --- | --- |
> | DietNeRF | 11.85 | 0.633 | 0.314 | 20.63 | 0.778 | 0.201 |
> | RegNeRF | 18.89 | 0.745 | 0.190 | 22.20 | 0.841 | 0.117 |
> | FreeNeRF | 19.92 | 0.787 | 0.182 | 23.25 | 0.844 | 0.131 |
> | 3DGS | 17.65 | 0.816 | 0.146 | 20.33 | 0.776 | 0.223 |
> | FSGS | 17.34 | 0.818 | 0.169 | 21.55 | 0.880 | 0.127 |
> | DNGaussian | 18.91 | 0.790 | 0.176 | 22.10 | 0.851 | 0.148 |
> | CoR-GS | 19.21 | 0.853 | **0.119** | 24.51 | 0.917 | **0.068** |
> | DropGaussian | 20.29 | 0.863 | 0.122 | 24.57 | 0.919 | 0.072 |
> | **D²GS (Ours)** | **21.25** | **0.865** | 0.121 | **25.25** | **0.920** | 0.071 |
>
> > **2.** Regarding typos in the LPIPS highlighting and the AVGE naming.
>
> We thank the reviewer for the careful reading and apologize for the spelling and annotation errors in the paper. First, in Table 1 for LLFF with 3 views at one quarter resolution, CoR-GS indeed achieves the best LPIPS of 0.250 and should be highlighted in red. We have corrected this in the revised version. Second, the mixed use of AVGS and AVGE is a naming mistake on our side. They actually refer to the same geometric error metric, and the correct notation is AVGE. In the revised manuscript, we have unified the notation to AVGE throughout the text and all tables to avoid ambiguity.

---

> ### Author Response · Authors · 2025-11-25
> **Response to Reviewer kaRU (Part 2/3)**
>
> > **3.** Regarding the comment that the method is engineering-oriented and has limited novelty.
>
> First, the core problem we aim to address is the spatially imbalanced failure modes of 3DGS under sparse views. We first systematically characterize this problem, and then design the most direct mechanisms around this characterization. Specifically, in the context of sparse-view 3DGS, we explicitly diagnose two typical failure modes: first, due to view bias, near-field Gaussians become overly dense and severely overfit; second, because of limited supervision and sparse sampling, far-field regions suffer from insufficient Gaussian coverage and thus underfitting. Unlike many prior works that mainly focus on pixel-space metrics or individual regularizers, we root these failure modes directly in the structure of the Gaussian distribution itself. This observation, and the underlying scientific problem it reveals, are of substantial importance in their own right.
>
> Second, we deliberately use simple methods to address this problem, which makes the solution more general and easier to summarize and transfer. We propose two mechanisms: one acting at the level of Gaussian primitives, and the other at the level of the training loss. The first is the depth–density guided dropout module, DD-Drop. Its core is not to add yet another heuristic point-dropping rule, but to incorporate depth and local density—the two geometric quantities that most directly capture view bias and redundant clustering in 3D scenes—into a unified, learnable, and decomposable dropout probability formulation. The depth term targets near-field bias, where viewpoints concentrate and nearby points are repeatedly observed and overfit. The density term targets local redundancy. The second mechanism is the distance-aware fidelity enhancement module, DAFE. Rather than simply adding another global weight, DAFE converts the depth prior provided by monocular depth estimation into an explicit reweighting of far-field pixels, correcting the spatial imbalance of supervision under sparse views at the loss level. Near-field pixels appear very frequently and produce large gradients, whereas far-field regions remain in a low-attention state for most of training. DAFE models this phenomenon as a depth bias in loss space and uses a soft mask to consistently assign slightly higher optimization priority to distant regions. In contrast to works that tightly embed depth priors into network architectures, we intentionally treat depth as a weak constraint and a soft reweighting signal. It does not impose rigid geometric constraints, yet it systematically improves far-field reconstruction quality.
>
> Overall, our contribution does not lie in introducing highly complex modules, but in three aspects: first, we provide a systematic analysis of failure mechanisms in sparse-view 3DGS at the level of Gaussian distributions; second, we propose a depth–density collaborative regularization mechanism that directly targets the core issue of misallocated Gaussian capacity over depth; third, through the simple yet structurally clear combination of DD-Drop and DAFE, we address this problem simultaneously in geometry space and in optimization space.

---

> ### Author Response · Authors · 2025-11-25
> **Response to Reviewer kaRU (Part 3/3)**
>
> > **4.** Regarding the observation that parameters in Table 5 seem to have little influence.
>
> We thank the reviewer for raising this point. The ablation experiments in Table 5 show that, after DD-Drop and DAFE are enabled, varying hyperparameters within a reasonable range leads to relatively small changes in PSNR, SSIM, and LPIPS. This does not mean that these parameters are unimportant or that the modules are ineffective. Instead, it indicates that our design behaves as a smooth and robust regularization mechanism, rather than a brittle, switch-like one.
>
> Our core contribution here is to explicitly introduce local density and depth, the two geometric quantities that most directly reflect sparse-view bias, into the dropout mechanism of Gaussian primitives. The joint depth–density signal determines which Gaussians should be dropped first and which should be preserved or strengthened. The fact that the results are not very sensitive to parameter changes suggests that these two factors operate stably and effectively over a fairly wide range of settings. Mechanistically, this low sensitivity to hyperparameters comes from the way DD-Drop and DAFE are designed: they both act through continuous modulation rather than a simple on/off switch.
>
> First, in DD-Drop, depth and density are combined as a weighted sum to form a single dropout score that controls the probability of dropping each Gaussian. This is a soft dropout strategy: adjusting the weights only smoothly shifts the emphasis between “more depth-driven” and “more density-driven,” rather than completely flipping which Gaussians are kept or discarded. Second, the dropout rate is constrained to lie within the interval [rmin⁡,rmax⁡] and changes continuously over the course of training, which prevents sudden oscillations such as aggressive over-pruning or almost no pruning at certain stages. Third, DAFE uses a depth-based soft mask and a loss reweighting with bounded magnitude, adding only a mild depth bias on top of the original supervision rather than drastically altering the optimization objective itself. Therefore, as long as the hyperparameters vary within a reasonable range, the overall training dynamics remain similar, and the final PSNR / SSIM / LPIPS curves change smoothly and stably, rather than exhibiting extremely fragile, parameter-sensitive behavior. If we completely remove a module (for example, by turning off DD-Drop or DAFE), performance clearly degrades. However, when these modules are enabled and their parameters lie in a reasonable range, the method is insensitive to the exact numerical values and consistently outperforms DropGaussian.
>
> We regard this as a positive property. The small performance variations under parameter changes in Table 5 show that depth and density signals are intrinsically meaningful for regularizing Gaussians and do not rely on extreme settings to be effective. They also show that the method is not highly sensitive to hyperparameters and has good robustness and practical usability. Our goal in these experiments is not to search for a single optimal weight, but to present the behavior under different choices. The fact that performance is stable across settings supports the effectiveness and robustness of our approach, which is beneficial for real-world use.

---

### Official Review · Reviewer_6fXt · 2025-10-30

**Soundness:** 2
**Presentation:** 3
**Contribution:** 2
**Rating:** 6
**Confidence:** 4

**Summary:**

This paper studies the instability of 3D Gaussian Splatting (3DGS) under sparse-view settings. The authors identify two common issues: overfitting in near-camera regions and underfitting in distant areas. To address them, they propose D2GS, which combines a depth-and-density guided dropout (DD-Drop) and a distance-aware fidelity enhancement (DAFE) module. The first adaptively suppresses redundant near-field Gaussians, while the second improves far-field reconstruction using depth-guided supervision. The paper also introduces a new evaluation metric, Inter-Model Robustness (IMR), based on the Wasserstein distance between Gaussian mixtures, to measure training stability. Experiments on LLFF and MipNeRF360 show consistent performance gains over existing NeRF- and 3DGS-based baselines.

**Strengths:**

1. The paper presents a clear and well-supported motivation, combining both visual analysis and quantitative evidence. The proposed method is technically simple yet effective, and the introduction of the IMR metric provides an interesting and complementary perspective on model robustness beyond standard reconstruction metrics.

2. The experimental evaluation is comprehensive, showing consistent quantitative improvements, strong qualitative comparisons, and detailed ablation studies. The additional analysis on hyperparameter sensitivity and the influence of different depth estimation models further reinforces the empirical soundness of the work.

**Weaknesses:**

1. The interpretability and practical utility of the IMR metric for model selection or training guidance remain limited, as its absolute values are still relatively high and its correlation with visual or perceptual quality is not fully established.
2. The method appears particularly effective for removing artifacts in unbounded scenes with significant depth variation, but its performance on object-centric datasets such as NeRF-Synthetic or DTU remains unclear.

**Questions:**

See weakness

---

> ### Author Response · Authors · 2025-11-25
> **Response to Reviewer 6fXt**
>
> > **1.** Regarding the design of the IMR metric.
>
> We thank the reviewer for their comments on IMR. As existing metrics such as PSNR, SSIM, and LPIPS already provide a fairly comprehensive evaluation of visual and perceptual quality, the original intention behind our proposed Inter-Model Robustness (IMR) metric is not to replace these standard metrics, but rather to characterize model stability from a complementary perspective—namely, by examining the behavior of Gaussian distributions obtained from multiple independently trained models.
>
> Under sparse-view conditions, even with a fixed scene and unified training configuration, existing 3DGS-based methods still exhibit significant performance degradation and instability: under the same input views and hyperparameters, they can yield substantially different 3D Gaussian models. In the rendered results, this instability manifests as significantly larger variance in PSNR, inconsistent background structures, and unreliable artifacts such as floating blobs. We believe that focusing solely on “how high PSNR/SSIM a single run can achieve” is insufficient; it is also essential to consider, from the perspective of a family of models, whether the learned 3D Gaussian distributions are stable under identical input conditions. This is precisely the motivation for proposing IMR as a distribution-level stability metric: it does not attempt to reinvent a visual quality metric, but instead quantifies the robustness and consistency of the 3D representation, complementing image-space metrics such as PSNR.
>
> From the perspective of image quality, in the LLFF 3-view setting, DropGaussian achieves an IMR of 3.205, whereas our method reaches 3.039 (where a lower value indicates better cross-model consistency). To more directly link IMR to visual outcomes, we conduct five independent training runs on the trex scene and visualize the corresponding renderings; these results are included in **Figure 7** of the revised version. Under identical training configurations, DropGaussian's renderings not only exhibit more overall artifacts and lower detail quality, but also show markedly larger differences across runs; in contrast, our method delivers more stable and higher-quality reconstructions across multiple trainings. This case study provides empirical evidence to some extent that IMR is meaningfully correlated with visual and perceptual quality in practice: lower IMR tends to correspond to more stable renderings and fewer artifacts.
>
> ---
>
> > **2.** Regarding performance on object-centric datasets.
>
> As you suggested, we add experiments on the DTU dataset, which contains single-object scenes with complex geometry and high-frequency details. And we also include these results in Table 2 and 8 in the revised version.
> As shown in the following Table, $D^2$GS achieves the best PSNR and SSIM among all NeRF-based and 3DGS-based baselines for both 3-view and 6-view settings, while maintaining competitive LPIPS. In the 3-view setting, D2GS improves PSNR by roughly 1–2 dB over DropGaussian and CoR-GS, achieves consistently higher SSIM, and maintains comparable or slightly better LPIPS. In the 6-view setting, it similarly yields about 0.7 dB PSNR gain with higher SSIM and a small improvement in LPIPS. Qualitatively, DropGaussian’s purely random drop strategy sometimes removes informative high-frequency Gaussians on the object surface, leading to oversmoothed textures and residual artifacts near boundaries, whereas $D^2$GS, guided by depth and density statistics, better preserves these details while still suppressing redundant near-field Gaussians.
>
> | Methods | PSNR (3-views) | SSIM (3-views) | LPIPS (3-views) | PSNR (6-views) | SSIM (6-views) | LPIPS (6-views) |
> | --- | --- | --- | --- | --- | --- | --- |
> | DietNeRF | 11.85 | 0.633 | 0.314 | 20.63 | 0.778 | 0.201 |
> | RegNeRF | 18.89 | 0.745 | 0.190 | 22.20 | 0.841 | 0.117 |
> | FreeNeRF | 19.92 | 0.787 | 0.182 | 23.25 | 0.844 | 0.131 |
> | 3DGS | 17.65 | 0.816 | 0.146 | 20.33 | 0.776 | 0.223 |
> | FSGS | 17.34 | 0.818 | 0.169 | 21.55 | 0.880 | 0.127 |
> | DNGaussian | 18.91 | 0.790 | 0.176 | 22.10 | 0.851 | 0.148 |
> | CoR-GS | 19.21 | 0.853 | **0.119** | 24.51 | 0.917 | **0.068** |
> | DropGaussian | 20.29 | 0.863 | 0.122 | 24.57 | 0.919 | 0.072 |
> | **D²GS (Ours)** | **21.25** | **0.865** | 0.121 | **25.25** | **0.920** | 0.071 |

---

### Official Review · Reviewer_TLEw · 2025-10-31

**Soundness:** 3
**Presentation:** 3
**Contribution:** 3
**Rating:** 6
**Confidence:** 4

**Summary:**

This paper presents D$^2$GS, a depth- and density-guided framework designed to improve the stability and accuracy of sparse-view 3D Gaussian Splatting (3DGS). The authors first identify two characteristic failure modes in sparse-view 3DGS: overfitting in near-field regions and underfitting in far-field regions. To address these issues, the paper introduces two main modules:
(1) Depth-and-Density Guided Dropout (DD-Drop), which regularizes Gaussian primitives using both local and global depth-density statistics to selectively drop overfitted points, and
(2) Distance-Aware Fidelity Enhancement (DAFE), which applies stronger supervision to underfitted far-field regions using monocular depth-based masks.
Additionally, the authors propose a novel evaluation metric called Inter-Model Robustness (IMR), derived from Wasserstein and Optimal Transport theory, to measure stability across independently trained models. Experiments on LLFF and Mip-NeRF360 datasets demonstrate consistent improvements in PSNR, SSIM, and robustness compared with recent 3DGS-based baselines such as CoR-GS and DropGaussian.

**Strengths:**

1. The paper is well written and clearly organized.

2. The proposed depth- and density-based method is supported by empirical observations, and the overall design is reasonable and novel.

3. The paper also proposes an IMR metric for evaluating the stability of 3DGS training, which provides a valuable new perspective for assessing robustness in sparse-view 3D reconstruction.

**Weaknesses:**

1. The method relies heavily on manually designed depth thresholds, which may limit its adaptability to scenes with large domain variation. It would be desirable for the authors to include comparisons on datasets with greater domain diversity.

2. Compared to a simpler baseline such as DropGaussian, this method introduces several additional heuristics and tunable parameters, but the performance gains appear relatively limited. Beyond analyzing the Gaussian distribution differences, more mechanistic experiments are needed to explain the underlying reason for the improvement.

3. The observation in Figure 1 regarding the depth distribution of Gaussian primitives is presented as a single example and lacks statistical analysis, making it insufficient to demonstrate the generality of the phenomenon.

4. The method relies on monocular depth estimation to generate masks and compute drop ratios; however, it does not account for potential inconsistencies between monocular depth maps across multiple input views, which could cause conflicting depth values and adversely affect Gaussian dropout and reconstruction quality.

5. The paper lacks discussion on feed-forward sparse-view 3DGS methods[1][2][3]. There is an important branch of literature addressing sparse-view reconstruction with general 3DGS frameworks that has not been discussed, resulting in incomplete coverage in the related works.

[1]pixelsplat: 3d gaussian splats from image pairs for scalable generalizable 3d reconstruction.

[2]Mvsplat: Efficient 3d gaussian splatting from sparse multi-view images.

[3]Hisplat: Hierarchical 3d gaussian splatting for generalizable sparse-view reconstruction.

**Questions:**

See the weaknesses.

---

> ### Author Response · Authors · 2025-11-25
> **Response to Reviewer TLEw (Part 1/3)**
>
> > **1.** Manually designed depth thresholds and domain diversity.
>
> Our depth thresholds are relative and scene-adaptive, not fixed global values. This means that the thresholds depend on the depth distribution of each scene, allowing the method to automatically adapt to the typical depth range and scale of the scene. In our current implementation, we do not use any fixed absolute depth values shared across scenes. The near / mid / far bands are defined via per-scene depth quantiles (first and second tertiles of the depth histogram). For example, in the LLFF "flower" sub-dataset, the foreground Gaussians account for 17.9%, the midground for 62.9%, and the background for 19.2%. In contrast, for the "horns" scene, the foreground Gaussians make up 68.3%, the midground 30.3%, and the background only 1.4%. This demonstrates how the depth bands adapt to the unique distribution characteristics of each scene, ensuring that the thresholds are scene-specific and not fixed across different environments. Similarly, in DAFE, the depth mask threshold $\tau$ is applied to normalized depth maps $D(x, y) / D_{\max}$, so it effectively selects “the farthest X\% pixels” rather than using a fixed metric depth. As a result, the same hyper-parameters can be applied across scenes with very different absolute depth ranges, and the notion of “far-field” is always defined relative to the current scene.
>
> Additional experiments on a more diverse dataset DTU. Following your suggestion, we have additionally evaluated D²GS on the DTU dataset, which is object-centric and has quite different geometry and depth characteristics compared to LLFF and MipNeRF360. Under the same sparse-view protocol, the results of different methods are as follows.
>
> | Methods | PSNR (3-views) | SSIM (3-views) | LPIPS (3-views) | PSNR (6-views) | SSIM (6-views) | LPIPS (6-views) |
> | --- | --- | --- | --- | --- | --- | --- |
> | DietNeRF | 11.85 | 0.633 | 0.314 | 20.63 | 0.778 | 0.201 |
> | RegNeRF | 18.89 | 0.745 | 0.190 | 22.20 | 0.841 | 0.117 |
> | FreeNeRF | 19.92 | 0.787 | 0.182 | 23.25 | 0.844 | 0.131 |
> | 3DGS | 17.65 | 0.816 | 0.146 | 20.33 | 0.776 | 0.223 |
> | FSGS | 17.34 | 0.818 | 0.169 | 21.55 | 0.880 | 0.127 |
> | DNGaussian | 18.91 | 0.790 | 0.176 | 22.10 | 0.851 | 0.148 |
> | CoR-GS | 19.21 | 0.853 | **0.119** | 24.51 | 0.917 | **0.068** |
> | DropGaussian | 20.29 | 0.863 | 0.122 | 24.57 | 0.919 | 0.072 |
> | **D²GS (Ours)** | **21.25** | **0.865** | 0.121 | **25.25** | **0.920** | 0.071 |
>
> We have added these DTU results in the revised manuscript in Table 2 and8 to better demonstrate cross-dataset generalization.

---

> ### Author Response · Authors · 2025-11-25
> **Response to Reviewer TLEw (Part 2/3)**
>
> > **2.** Extra parameters vs. DropGaussian.
>
> We appreciate the reviewer’s comments regarding the additional heuristics and hyper-parameters introduced on top of DropGaussian. We would like to clarify that, although $D^2$GS does introduce several new components, each of them is directly motivated by two concrete failure modes we consistently observe in sparse-view 3DGS: (i) overfitting caused by excessively dense Gaussians near the cameras, and (ii) underfitting in distant regions due to insufficient Gaussian coverage. Our goal is not to stack arbitrary tricks, but to systematically address these two issues.
>
> From a mechanistic perspective, the added hyper-parameters are designed to implement depth–density–guided regularization in a controlled manner. In the DD-Drop module, $\omega_{\text{depth}}$ and $\omega_{\text{density}}$ are used to construct a dropout score that combines normalized depth and local density: the depth term primarily suppresses Gaussians that are very close to the cameras, thereby counteracting the inherent near-field bias under sparse viewpoints, while the density term penalizes overly dense clusters in texture-rich regions to avoid local redundancy and overfitting. The dropout schedule parameters $r_{\min}$ and $r_{\max}$ define the lower and upper bounds of the time-varying dropout rate: a smaller rate at early iterations helps preserve essential geometry and stabilize convergence when the Gaussian set is still being formed, whereas gradually increasing the rate toward $r_{\max}$ in later stages plays the role of structured pruning, effectively regularizing redundant Gaussians once the representation has become sufficiently dense. Together with the component-wise ablations, these results indicate that each parameter has a clear and interpretable function in the pipeline, and that the full model relies on these targeted designs to systematically suppress near-field overfitting while enhancing far-field fidelity, rather than depending on arbitrary tuning.
>
> Compared to DropGaussian, our design is driven by two main observations, as illustrated in Figure 1. From the perspective of Gaussian primitives, DropGaussian tends to place too many Gaussians in near-field regions, which leads to overfitting, while allocating too few Gaussians to far-field regions, which leads to underfitting. From the perspective of the rendered RGB images, this unbalanced Gaussian distribution translates into noticeably degraded appearance in distant areas and lower PSNR, indicating that purely random dropout indeed struggles to handle sparse-view bias in a principled way.
>
> Quantitatively, as reported in Tables 1 and 2 in the revised version, $D^2$GS consistently improves over DropGaussian and other 3DGS baselines across multiple settings. On LLFF 3-view at $1/8$ resolution, $D^2$GS reaches 21.35 dB PSNR and 0.746 SSIM, compared to 20.76 dB and 0.713 for DropGaussian, corresponding to a gain of 0.59 dB in PSNR. At the same time, LPIPS is reduced from 0.200 to 0.179 (≈10.5% relative reduction), and AVGE decreases from 0.097 to 0.087 (≈10.3% reduction). This improvement far exceeds that of other methods. Apart from this, the IMR metric also demonstrates that the model is more robust, which is itself an improvement in performance.
>
> In the revised version, we include results on the DTU dataset in Table 2 and 8, which contains object-centric scenes with complex geometry and rich high-frequency details. As shown in the above Table in Response to Reviewer TLEw Part 1, $D^2$GS achieves the best PSNR and SSIM among all NeRF-based and 3DGS-based baselines for both 3-view and 6-view settings, while maintaining competitive LPIPS. In the 3-view setting, D2GS improves PSNR by roughly 1–2 dB over DropGaussian and CoR-GS, achieves consistently higher SSIM, and maintains comparable or slightly better LPIPS. In the 6-view setting, it similarly yields about 0.7 dB PSNR gain with higher SSIM and a small improvement in LPIPS. Qualitatively, DropGaussian’s purely random drop strategy sometimes removes informative high-frequency Gaussians on the object surface, leading to oversmoothed textures and residual artifacts near boundaries, whereas $D^2$GS, guided by depth and density statistics, better preserves these details while still suppressing redundant near-field Gaussians.

---

> ### Author Response · Authors · 2025-11-25
> **Response to Reviewer TLEw (Part 3/3)**
>
> > **3.** Generality of the depth-distribution phenomenon and statistical analysis.
>
> As you suggested, we extend analysis in following Table and also add in Table 10 in the revised version to include multiple representative scenes. Specifically, we evaluate three scenes (horns, leaves, orchids) from LLFF and an average LLFF case, and compare sparse-view models (ours, DropGaussian, and CoR-GS) against a dense-view reference model. For each scene, we run each model 10 times to ensure robust results, and we compute the average proportion of Gaussians in the near and far fields. Across all scenes, we consistently observe two key phenomena: near-field overpopulation and far-field underpopulation in the sparse-view models. This imbalance is due to sparse supervision, where most Gaussians crowd into the near field while distant regions are insufficiently represented. Our method effectively addresses this issue by redistributing the Gaussians more evenly across both the near and far fields, aligning more closely with the dense-view reference. In contrast to DropGaussian, which shows extreme overpopulation in near-field regions and severe underpopulation in the far field, our method provides a more balanced distribution of Gaussians, improving far-field representation while avoiding overfitting in near-field regions. These results are stable across all tested scenes, demonstrating that the depth-distribution imbalance is a general phenomenon, and that our approach consistently improves Gaussian allocation, making the model more robust and better aligned with the true scene structure. This highlights the effectiveness of our method in correcting depth-related biases and improving overall 3D representation quality.
>
> | Methods                      | horns near | horns far | leaves near | leaves far | orchids near | orchids far | LLFF near | LLFF far |
> |------------------------------|------------|-----------|-------------|------------|--------------|-------------|-----------|----------|
> | CoR-GS (Sparse-view)         | 61.16%     | 38.84%    | 51.29%      | 48.71%     | 92.19%       | 7.81%       | 55.45%    | 44.55%   |
> | DropGaussian (Sparse-view)   | 77.40%     | 22.60%    | 43.46%      | 56.54%     | 92.51%       | 7.49%       | 55.93%    | 44.07%   |
> | D²GS (Sparse-view)           | 59.02%     | 40.98%    | 34.18%      | 65.82%     | 90.63%       | 9.37%       | 51.49%    | 48.51%   |
> | Dense-view                   | 47.07%     | 52.93%    | 24.55%      | 75.45%     | 88.61%       | 11.39%      | 51.55%    | 48.45%   |
>
> ---
>
> > **4.** The use of monocular depth estimation.
>
> We use depth estimation models that produce relative depth, and we only use the per-view relative depth as a loss-weighting signal to increase the weight of background regions, rather than as a strict object geometric constraint, so our method does not strongly depend on the depth model. DAFE functions as a soft attention prior, simply encouraging the model to focus on distant regions, rather than as a fragile hard supervision signal, and whether the depth maps are consistent or not has very little impact on far-field optimization.
>
> Our method is robust to different depth estimators. We compare three different monocular depth networks (MiDaS / DPT / DepthAnything) in Table 9 and observe that for all of them, $D^2$GS consistently improves over DropGaussian. The performance differences between depth models are much smaller than the performance gap between $D^2$GS and DropGaussian, which indicates that the method is reasonably robust to the choice of depth estimator as well as the noise or inconsistencies it may introduce.
>
> ---
>
> > **5.** Discussion of feed-forward sparse-view 3DGS methods.
>
> As you suggested, we add a discussion of these feed-forward methods in the Related Work section of our revised version.
> With more efficient 3DGS frameworks, recent methods improve scene understanding via pseudo-view generation, address sparse initialization with additional priors, and mitigate overfitting to training views. Recently, some feed-forward methods further advance sparse‐view NVS: PixelSplat predicts 3D Gaussian parameters directly from images, MVSplat incorporates multi-view stereo cues to improve depth reliability under sparse inputs, and HiSplat adopts a hierarchical Gaussian representation to enhance geometric detail and view consistency.

---

> > ### Comment · Reviewer_TLEw · 2025-11-28
> >
> > Thanks for your response. I think my core concerns are addressed, and I support the acceptance of this paper.

---

> > > ### Author Response · Authors · 2025-11-28
> > > **Response to Reviewer TLEw**
> > >
> > > Thank you for your insightful comments and appreciation of our work and rebuttal. It is good to see that our comments could address your concerns. We will do our best to improve the final version of our paper based on your valuable suggestions.

---

### Author Response · Authors · 2025-11-25
**Summary**

We thank all reviewers for their positive feedback:
- The motivation is clear, empirically supported, and the proposed method is reasonable and technically simple yet effective (TLEw, 6fXt, kaRU).
- The introduced IMR metric offers a valuable new perspective for evaluating training stability and model robustness (TLEw, 6fXt).
- Experiments are comprehensive, show consistent improvements across datasets, and include strong qualitative comparisons and detailed ablations (6fXt, kaRU).
- The paper is well written, clearly organized, and easy to follow (TLEw, kaRU).

We address the raised concerns as follows.

---

### Meta-Review · Area_Chair_J52s · 2026-01-03

**Summary:**

All the reviewers lean towards accepting this work even though many concerns are raised. The key concerns are:
- Limited Generality & Robustness: The method relies on manually tuned depth thresholds, raising concerns about its adaptability to diverse scenes (e.g., object-centric datasets like DTU).
- Insufficient Justification & Analysis: Compared to simpler baselines, the performance gain appears marginal relative to its added complexity. The paper lacks mechanistic experiments to explain why it works and statistical validation for its core observations.
- Potential Technical Flaw: The approach depends on monocular depth estimation but ignores inconsistencies across views, which could harm the dropout process and final quality.

The authors well addressed these concerns, including:
- The thresholds can adapt to diverse scenes through "normalized depth map".
- Experiments on DTU are provided, and the proposed method overperform DropGS by 0.98 in PSNR.
- Table 9 explains the influence about depth inconsistencies.

Taking the comments into consideration, this paper is recommended for acceptance.

**Reviewer Concerns:**

Please see above.

**Reviewer Scores:**

Please see above.

---

### Decision · Program_Chairs · 2026-01-26

Accept (Poster)